# Establishment and Molecular Characterization of Two Patient-Derived Pancreatic Ductal Adenocarcinoma Cell Lines as Preclinical Models for Treatment Response

**DOI:** 10.3390/cells12040587

**Published:** 2023-02-11

**Authors:** Rüdiger Braun, Olha Lapshyna, Jessica Watzelt, Maren Drenckhan, Axel Künstner, Benedikt Färber, Ahmed Ahmed Mohammed Hael, Louisa Bolm, Kim Christin Honselmann, Björn Konukiewitz, Darko Castven, Malte Spielmann, Sivahari Prasad Gorantla, Hauke Busch, Jens-Uwe Marquardt, Tobias Keck, Ulrich Friedrich Wellner, Hendrik Ungefroren

**Affiliations:** 1Department of Surgery, University Medical Center Schleswig-Holstein, Campus Lübeck, 23538 Lübeck, Germany; 2Medical Systems Biology Group, Lübeck Institute of Experimental Dermatology, University of Lübeck, 23538 Lübeck, Germany; 3Institute for Cardiogenetics, University of Lübeck, 23538 Lübeck, Germany; 4First Medical Department, University Medical Center Schleswig-Holstein, Campus Lübeck, Ratzeburger Allee 160, 23538 Lübeck, Germany; 5Institute of Pathology, University Medical Center Schleswig-Holstein, Campus Kiel, 24105 Kiel, Germany; 6Institute of Human Genetics, University of Lübeck, 23538 Lübeck, Germany; 7Department of Hematology and Oncology, University Medical Center Schleswig-Holstein, Campus Lübeck, 23538 Lübeck, Germany

**Keywords:** pancreatic cancer, primary cell line, patient-derived cancer model, precision medicine

## Abstract

The prognosis of pancreatic ductal adenocarcinoma (PDAC) is exceedingly poor. Although surgical resection is the only curative treatment option, multimodal treatment is of the utmost importance, as only about 20% of tumors are primarily resectable at the time of diagnosis. The choice of chemotherapeutic treatment regimens involving gemcitabine and FOLFIRINOX is currently solely based on the patient’s performance status, but, ideally, it should be based on the tumors’ individual biology. We established two novel patient-derived primary cell lines from surgical PDAC specimens. LuPanc-1 and LuPanc-2 were derived from a pT3, pN1, G2 and a pT3, pN2, G3 tumor, respectively, and the clinical follow-up was fully annotated. STR-genotyping revealed a unique profile for both cell lines. The population doubling time of LuPanc-2 was substantially longer than that of LuPanc-1 (84 vs. 44 h). Both cell lines exhibited a typical epithelial morphology and expressed moderate levels of CK7 and E-cadherin. LuPanc-1, but not LuPanc-2, co-expressed E-cadherin and vimentin at the single-cell level, suggesting a mixed epithelial-mesenchymal differentiation. LuPanc-1 had a missense mutation (p.R282W) and LuPanc-2 had a frameshift deletion (p.P89X) in *TP53*. *BRCA2* was nonsense-mutated (p.Q780*) and *CREBBP* was missense-mutated (p.P279R) in LuPanc-1. *CDKN2A* was missense-mutated (p.H83Y) in LuPanc-2. Notably, only LuPanc-2 harbored a partial or complete deletion of *DPC4*. LuPanc-1 cells exhibited high basal and transforming growth factor (TGF)-β1-induced migratory activity in real-time cell migration assays, while LuPanc-2 was refractory. Both LuPanc-1 and LuPanc-2 cells responded to treatment with TGF-β1 with the activation of SMAD2; however, only LuPanc-1 cells were able to induce TGF-β1 target genes, which is consistent with the absence of *DPC4* in LuPanc-2 cells. Both cell lines were able to form spheres in a semi-solid medium and in cell viability assays, LuPanc-1 cells were more sensitive than LuPanc-2 cells to treatment with gemcitabine and FOLFIRINOX. In summary, both patient-derived cell lines show distinct molecular phenotypes reflecting their individual tumor biology, with a unique clinical annotation of the respective patients. These preclinical ex vivo models can be further explored for potential new treatment strategies and might help in developing personalized (targeted) therapy regimens.

## 1. Introduction

Pancreatic cancer is currently the fourth most common cause of cancer-related deaths and is predicted to be the second leading cause of cancer-related mortality by the year 2030 [1]. The majority of malignant neoplasms of the pancreas are ductal adenocarcinomas (PDAC) [2], and the prognosis of PDAC is exceedingly poor due to several reasons. Most PDACs are notably resistant to conventional (radio)chemotherapy.

Surgical resection is the only curative treatment option for pancreatic cancer. However, only 10–20% of patients present with resectable tumors at the time of diagnosis, 30–40% present with a borderline resectable tumor, and 50–60% present with metastatic disease [2]. There is growing evidence for a better oncological outcome for patients with borderline resectable tumors after neoadjuvant treatment [3,4]. A higher proportion of R0 resections can be achieved in patients with resectable and borderline resectable PDAC after neoadjuvant treatment [5].

The adjuvant treatment of patients with resectable PDAC by chemotherapy results in better overall survival. Modified combination therapy consisting of 5-fluorouracil, leucovorin, irinotecan, and oxaliplatin (mFOLFIRINOX) resulted in a significantly better median overall survival of 54.4 months compared to that of gemcitabine alone (35.0 months) [6]. However, the progress in the treatment of patients with advanced-stage disease has been modest. Combination therapies such as mFOLFIRINOX or gemcitabine with albumin-bound (nab)-paclitaxel only result in marginal improvements in the overall survival in these patients [7,8].

Overall, the 5-year survival rate of PDAC patients remains as low as 5–8% [9]. The selection of mFOLFIRINOX or gemcitabine (with nab-paclitaxel) is currently based on the patient’s performance status and comorbidities, whereas the individual tumor biology is not considered. Multiple studies were aimed at establishing molecular profiles in PDACs [10,11,12]. As reviewed by Nevala-Plagemann et al., comprehensive genomic profiling might facilitate the identification of subsets of patients with actionable alterations who can potentially benefit from new targeted therapies [13]. For instance, genomic loss in the cancer cells of *CDKN2A* (encoding the cyclin-dependent kinase inhibitor p16^INK4^) and *DPC4*, encoding SMAD4, a central mediator of transforming growth factor (TGF)-β signaling, has been implicated in modifying the response to gemcitabine or 5-FU [14,15,16]. However, targeted therapies for PDAC patients have not become an integral part of clinical routine practice so far [17]. Taken together, there is an unmet clinical need (i) to identify biomarkers that are predictive of the response to treatment and allow for personalized therapy and (ii) to find alternative, targeted treatment options for patients who do not respond to treatment regimens that are currently used in clinical practice.

Preclinical models of PDAC are essential for drug development and the identification of predictive biomarkers. Primary, patient-derived cell lines are an indispensable tool among ex vivo models. Primary cell lines can provide valuable information on the phenotype and genotype of individual tumors, allow for functional assays of the individual tumor biology, and be used for the development of personalized treatment strategies by enabling individualized therapeutic drug testing ex vivo. However, the efficient ex vivo expansion of human carcinomas in tissue culture, especially that of PDAC, remains technically challenging. The Cancer Cell Line Encyclopedia (CCLE) from the Broad Institute comprises 60 pancreatic cell lines. Nevertheless, the full spectrum of the complex and diverse tumor biology of PDAC is not reflected sufficiently by these cell lines, and the clinical annotation and oncological outcome of the respective patient are mostly missing.

Here, we describe two newly established patient-derived cell lines from surgical PDAC specimens named LuPanc-1 and LuPanc-2. Both cell lines were comprehensively characterized and tested with respect to the standard therapy regimens gemcitabine and FOLFIRINOX.

## 2. Materials and Methods

### 2.1. Tissue Collection, Cell Line Establishment, Fibroblast Depletion, and Generation of Single-Cell-Derived Clonal Cultures

The tissue was collected directly after the surgical resection of the tumor after approval by the University of Lübeck’s local ethics committee (# 16-281). The tissue was processed within 1 h after surgical resection and cut into 3–5 mm pieces using a scalpel. Pieces were dissociated using collagenase I (1000 U/mL; Gibco, Waltham, MA, USA) for 120 min at 37 °C. The cells were plated on tissue culture dishes in DMEM GlutaMAX^TM^ with 4.5 g/L glucose (Gibco) supplemented with 20% (*v*/*v*) fetal calf serum (FCS) and 100 μg/mL streptomycin/penicillin. The cells were cultured at 37 °C in a humidified incubator with 5% CO_2_. The cells began to form colonies after about 7 d. Subsequently, the medium was replaced every 3–4 d. The number of cocultured fibroblasts was periodically reduced by differential trypsinization. After reaching stable growth of the cultured tumor cells, fibroblasts were depleted by an immunomagnetic assay based on fibroblast-specific antigens (Anti-Fibroblast MicroBeads, human, Miltenyi Biotec, Bergisch Gladbach, Germany). Single-cell-derived clonal cultures were established from primary cultures by limited dilution, as previously described for the permanent PDAC cell line PANC-1 [18]. The cells were routinely checked for mycoplasma contamination by PCR (MycoScope PCR Detection Kit, Genlantis, San Diego, CA, USA). Cell line authentication was performed by short tandem repeat (STR) profiling using the PowerPlex^®^ 21 System (Promega, Madison, WI, USA), according to the manufacturer’s instructions.

### 2.2. Growth Curves

Growth curves for both cell lines were established by counting cells in three independent experiments every 24 h for 7 and 10 d for LuPanc-1 and LuPanc-2, respectively, after seeding 80,000 cells per well in a six-well plate. The final cell count was carried out after harvesting the cells using Accumax (PAN Biotech) in an improved Neubauer hemocytometer. The total cell number was determined by averaging the cell count in eight 1 mm squares. The cell numbers were normalized to the cell number initially counted 24 h after seeding.

### 2.3. Chemotherapy and Measurement of Cell Survival

To measure the sensitivity to chemotherapy, 6000 cells of LuPanc-1 and 8000 cells of LuPanc-2 growing in a log phase were seeded per well in 96-well plates, and chemotherapeutics were added 24 h after cell seeding. For gemcitabine treatment, gemcitabine (Sigma Aldrich, St. Louis, MO, USA) was added to a final concentration in a range of 2.5–320 nmol/L for LuPanc-1 and 10–12,800 nmol/L for LuPanc-2. For FOLFIRINOX treatment, 5-FU, irinotecan, leucovorin, and oxaliplatin were combined, and the final concentration ranged as follows: 5-FU (Medac) 0.005–426 µmol/L, irinotecan (Medac) 0.07 nmol/L–5.78 µmol/L, oxaliplatin (GRY) 0.062 nmol/L–4.96 µmol/L, and leucovorin (Sigma Aldrich) 0.078 nmol/L–6.19 µmol/L. After 72 h, cell metabolism was measured by a Resazurin reduction assay. Resazurin (STEMCELL) was diluted in DPBS to a concentration of 10 mg/mL. This stock solution was further diluted in the medium, 20 µL was added per well to a final concentration of 40 µg/mL, and the cells were incubated for 2 h at 37 °C in a humidified incubator with 5% CO_2_. Extinction was quantified with a plate photometer, according to the manufacturer’s instructions. The excitation wavelength was set to 545 nm, and the emission wavelength was set to 600 nm. Each experiment was performed in triplicate and repeated three times.

### 2.4. Immunofluorescence

The cells were cultivated for 72 h, with subsequent fixation with 4% paraformaldehyde in chamber slides (Falcon^®^ Culture Slides; Corning, NY, USA and Sarstedt, Nümbrecht, Germany). The cells were permeabilized with 0.1% Triton-X and sequentially treated with goat serum, primary monoclonal mouse anti-human CK 7 antibody (DAKO Corporation, Clone OV-TL 12/30, 1:200 dilution), primary monoclonal rabbit anti-human E-cadherin antibody (Cell Signaling Technology, Frankfurt/Main, Germany, #3195, 1:200 dilution), primary monoclonal mouse anti-human vimentin antibody (Abcam, Cambridge, UK, V1-RE/1, ab3974,1:200 dilution), anti-Mouse Alexa Fluor^®^ 594 secondary antibody (Abcam, #150116, 1:500 dilution), and anti-Rabbit Alexa Fluor^®^ 488 secondary antibody (Abcam, #150077, 1:500 dilution). Appropriate negative controls were performed. Coverslips were mounted with Fluoromount-G medium including DAPI (Southern Biotech, Huntsville, AL, USA). The cells were visualized using a BZ-9000 (BIOREVO) fluorescent microscope (Keyence, Osaka, Japan) with a 20× PlanFluor EL NA 0.45 Ph1 objective lens under 0.285 s of exposure time for CK7 and 0.2 s of exposure time for E-cadherin and vimentin.

### 2.5. Immunoblotting

The Western blot procedure was performed as described in detail earlier [19]. Briefly, equal amounts of crude proteinaceous extracts from each cell line (three preparations harvested at different times during the continuous culture) were fractionated by SDS-PAGE on mini-PROTEAN TGX any-kD precast gels or TGX Stain-Free FastCast gels (BioRad, Munich, Germany) and blotted to PVDF membranes. The membranes were blocked with nonfat dry milk or bovine serum albumin and incubated with primary antibodies to E-cadherin (#610181, BD Transduction Laboratories, Heidelberg, Germany) or vimentin (clone V9, #V6630, Sigma-Aldrich, Steinheim, Germany), phospho-Smad2 (S465/S467) (#3101, Cell Signaling Technology, Frankfurt/Main, Germany), Smad2 (#1736-1, Epitomics, Burlingame, CA, USA), and GAPDH (Cell Signaling Technology) as loading controls.

### 2.6. Soft Agar/Matrigel-Based Sphere-Forming Assays

To determine the sphere-forming potential of Lü-PANC cells, Soft Agar- and Basement Membrane Extract (BME)-based assays were employed. For the Soft Agar-based assay, a total of 1000 viable single-cells were resuspended in 0.25% agar (Carl Roth, Karlsruhe, Germany) and added on top of the precoated 48-well plate with 1.3% agar containing DMEM with 20% FBS. For the sphere formation assay in BME, 1000 cells were resuspended in 50 µL DMEM with 20% FBS, mixed with 150 µL Cultrex (Bio-Techne, Minneapolis, MN, USA), and then plated on 48-well plates. During spheroid cultivation, the medium was replaced every 3 d. Sphere formation in both assays was monitored for 14 d. All experiments were performed in two independent biological replicates with three technical replicates each.

### 2.7. Migration Assays

The migratory/invasive activities of LuPanc-1 and LuPanc-2 were determined with xCELLigence^®^ technology (ACEA Biosciences, San Diego, supplied by OLS, Bremen, Germany), as outlined in detail in previous publications [19,20]. The lower side of the CIM plate-16 porous membrane was coated with a 1:1 mixture (*v*/*v*) of collagen I and collagen IV (30 μL) to facilitate the adherence of the cells and thus enhance the duration of the signal recording. Each well was loaded with 60,000 (LuPanc-1) or 80,000 (LuPanc-2) cells in standard growth medium (see above). Cells were allowed to settle in the laminar flow hood for 30 min at RT, after which the assay was started and run for 24 or 48 h. Some assays were performed in the absence or presence of recombinant human (rh) TGF-β1 (RELIATech GmbH, Wolfenbüttel, Germany) or the TGF-β type I receptor/ALK5 serine/threonine kinase inhibitor SB431542 (Merck, Darmstadt, Germany). Data acquisition (with signal recording every 15 min) and analysis were performed with the RTCA software (version 1.2, ACEA Biosciences).

### 2.8. Whole Exome and Sanger Sequencing

DNA was extracted from both cell lines using the AllPrep DNA/RNA/Protein Mini Kit (Qiagen, Hilden, Germany), according to the manufacturer’s instructions. Targets were enriched using the Agilent SureSelectXT kit, and libraries were prepared using the NEB Next Ultra DNA Library Prep Kit. Sequencing was performed on a NovaSeq 6000 using the v1.5 chemistry.

Sequencing data (paired-end fastq files) were trimmed (adapter and quality values) using fastp [21] (v0.23.0; minimum length, 50 bp; maximum unqualified bases, 30%; trim tail set to 1), and filtered reads were mapped against GRCh38 using bwa mem [22] (v0.7.15). Mappings were converted to BAM format using Picard Tools (v2.18.4), and postprocessing was performed following GATK’s [23] best practices. Briefly, mate-pair information was fixed, PCR duplicates were removed, and base quality recalibration was performed using Picard Tools, GATK (v4.2.4.1), and dbSNP v138 [24]. Single-nucleotide variants (SNVs) and short insertions and deletions (indels) were identified following Mutect2 (GATK) in the tumor-only mode, with gnomAD as a germline resource and the b38 exome panel from the 1000-genome project as a panel of normals. Variants were filtered and annotated using Variant Effect Predictor (VEP v103, GRCh38; adding CADD v1.6, dbNSFP v4.1a10, and GNOMAD r3.0 as additional annotations), and annotations were converted into MAF format using vcf2maf (V1.6.21) (http://doi.org/10.5281/ZENODO.593251 (accessed on 24 April 2022); coverage for each variant was extracted directly from the vcf INFO field. The top 20 frequently mutated genes (FLAGS) [25] and MUC genes were removed from further analysis, and the remaining somatic variants were filtered as follows: minimum coverage of 40, minimum variant allele frequency of 10%, and only variants with a frequency < 0.1% in 1000 genomes, gnomAD, or ExAC were considered for subsequent downstream analysis.

Copy number aberrations were detected using Control-FREEC (v11.4) [26], and sequencing coverage was estimated using mosdepth (v0.3.2) [27], resulting in 122x coverage for LuPanc1 (±93x) and 115x coverage for LuPanc-2 (±86x).

For the Sanger sequencing of the KRAS locus, the following primers were used (5′–3′): forward: CCTTATGTGTGACATGTTCT; reverse: ATGGTCCTGCACCAGTAATA (amplicon size: 218 bp). Sanger sequencing was carried out as described previously [28].

### 2.9. Quantitative RT-PCR Analysis (qPCR)

The total RNA was isolated from LuPanc-1 and LuPanc-2 cells with an RNAeasy kit (Qiagen, Hilden, Germany). A total of 2.5 μg of RNA of each sample was reverse transcribed (1 h, 37 °C) with 200 U of M-MLV Reverse Transcriptase and 2.5 μM random hexamers. Target gene expression was quantified by qPCR on an I-Cycler (BioRad, Munich, Germany) with Maxima SYBR Green Mastermix (Thermo Fisher Scientific, Waltham, MA, USA) and normalized to the expression of either TATA box-binding protein (TBP) or GAPDH. The PCR primer sequences for *DPC4*, *SMAD2*, *SERPINE1*, *SNAI2*, *SOX2, TBP* and *GAPDH* are given in Appendix A.

## 3. Results

### 3.1. Patient History and the Establishment of Patient-Derived Cell Lines

We established two patient-derived primary pancreatic cancer cell lines from PDACs (LuPanc-1 and LuPanc-2). LuPanc-1 cells were derived from a PDAC of a 64-year-old male patient who presented to our surgical department with a locally advanced tumor of the pancreatic tail with infiltration of the stomach. The patient underwent distal pancreatectomy, splenectomy, and simultaneous partial resection of the stomach. The histopathological assessment of the resected tumor revealed a moderately differentiated ductal adenocarcinoma of the pancreas (TNM classification: pT3,N1,L1,V1,Pn1,R0,G2). The patient received adjuvant chemotherapy with a reduced dose of FOLFIRINOX due to intolerance. The patient developed a local recurrence and peritoneal carcinomatosis 9 months after the surgical resection of the primary tumor and subsequently received palliative chemotherapy with gemcitabine/nab-paclitaxel.

LuPanc-2 cells were derived from a locally advanced PDAC of a 63-year-old female patient. The patient presented with a tumor located in the pancreatic corpus with local infiltration of the portal vein and celiac trunk. The patient underwent an extended multiorgan pancreatic corpus and tail resection (Appleby procedure) at our department. The histopathological assessment of the resected tumor revealed a poorly differentiated ductal adenocarcinoma of the pancreas (TNM classification: pT3,N2,L1,V1,Pn1,R0,G3). The patient received adjuvant chemotherapy with FOLFIRINOX but developed a retrogastric local recurrence 8 months after the resection of the primary tumor. The patient subsequently received palliative radiotherapy.

For both cell lines, cells were isolated immediately after the surgical resection of the tumor by mechanical and enzymatic dissociation of a representative specimen. After 7 days of cultivation, first, adherent colonies of epithelial tumor cells surrounded by fibroblasts appeared, as shown exemplarily for LuPanc-1 in Figure 1A. The number of cocultured fibroblasts was periodically reduced by differential trypsinization (Figure 1B). After reaching the stable growth of the cultured tumor cells, the fibroblasts were depleted by an immunomagnetic assay (Figure 1C). Both patient-derived PDAC cell lines showed a typical epithelial morphology. LuPanc-1 cells were small and grew in dense, large colonies, whereas LuPanc-2 cells were comparatively large and grew in small islets (Figure 1D,E). Both cell lines expressed moderate levels of cytokeratin 7 (CK7), as determined by immunofluorescent staining (Figure 1F–K).

### 3.2. Authentication of LuPanc-1 and LuPanc-2

Both established cell lines were authenticated by STR genotyping for 21 loci (Table 1). No identical STR profiles were found in the DSMZ STR database, indicating that both cell lines are novel [29]. We did not detect multiple peaks in the examined loci, which excludes cross-contamination by other cell lines. Contamination with mycoplasma was excluded using a PCR-based assay (data not shown).

### 3.3. Growth Kinetics and Epithelial-Mesenchymal Phenotyping

First, we determined the general growth behavior of both cell lines. The population doubling times of LuPanc-1 and LuPanc-2 cells were estimated to be 44 h and 84 h, respectively, as measured by manual cell counting, indicating that LuPanc-2 cells grew substantially slower than LuPanc-1 cells (Figure 2A). To further asses cellular growth, general metabolic activity was measured based on the Resazurin reduction assay. In line with the cell counting results, the metabolic activity was substantially lower in LuPanc-2 cells compared to that in LuPanc-1 cells. With increasing cell density, however, metabolic activities became more similar in both cell lines (Appendix A).

The aggressiveness of pancreatic cancer cells is associated with the acquisition of mesenchymal characteristics in comparison to those cells that are well-differentiated and epithelial-like [30]. Thus, we determined for both cell lines whether the cells are more epithelial or mesenchymal-like by measuring the protein levels of E-cadherin and vimentin by Western Blotting. Interestingly, LuPanc-1 cells expressed both E-cadherin and vimentin, while LuPanc-2 cells presented with similar levels of E-cadherin but lacked detectable levels of vimentin (Figure 2B). In separate immunoblots, LuPanc-1 cells were then compared to several permanent cell lines representing either the quasimesenchymal subtype—with a high expression of vimentin but only low (PANC-1) or undetectable (MIA PaCa-2) levels of -E-cadherin—or the epithelial subtype with strong E-cadherin but no vimentin expression (CAPAN-1 and COLO 357). From the data in Figure 2C, it became clear that LuPanc-1 cells cannot be attributed to one of these extreme phenotypes. Rather, they display a mixed phenotype combining both epithelial and mesenchymal traits. To reveal whether this is due to two different populations of cells, of which one expresses E-cadherin and the other expresses vimentin, or, alternatively, to the co-expression of both markers by the same individual cell, we performed double immunofluorescent staining of the E-cadherin and vimentin of LuPanc-1 cells. Interestingly, these stains show that individual LuPanc-1 cells indeed co-express both proteins but also appear to differ in their ratio of E-cadherin and vimentin expression (Figure 2D–G). To confirm this latter observation, we additionally generated single-cell-derived clonal cell lines from the parental LuPanc-1 culture by limited dilution. The immunoblot analysis of E-cadherin and vimentin expression in these clonal cultures mirrored the results from the immunofluorescence analyses. Specifically, all six single-cell-derived clones co-expressed both proteins, albeit with substantial quantitative differences, which were stronger for vimentin than for E-cadherin (Appendix A). The ratios of E-cadherin to vimentin were similar among the clones, except for clone 6F9, which displayed very strong vimentin expression and, hence, a low E-cadherin:vimentin ratio (Appendix A). Together, these data confirm vimentin expression by the tumor cells rather than by residual contaminating fibroblasts and suggest that LuPanc-1 cells undergo spontaneous EMT.

### 3.4. Genomic Landscape

We profiled the genomic landscape of both cell lines by whole-exome sequencing (WES). After the removal of commonly mutated genes without any effect, 135 and 76 mutations were detected in LuPanc-1 and LuPanc-2, respectively. Most of the mutations were missense mutations in both cell lines, followed by nonsense mutations (Table 2). Mutations in oncogenes and tumor suppressor genes, according to Vogelstein and coworkers [32], were detected as follows: *TP53* had a missense mutation in LuPanc-1 (p.R282W) and a frameshift deletion (p.P89X) in LuPanc-2 (Figure 3A). *BRCA2* was nonsense-mutated (p.Q780*) in LuPanc-1. *CDKN2A* (encoding p16^INK4^) had a missense mutation (p.H83Y) in LuPanc-2 and *CREBBP* had a missense mutation (p.P279R) in LuPanc-1. We further specifically analyzed *KRAS*, *DPC4*, *RNF43*, *ARID1A*, *TGFBR2*, *GNAS*, *RREB1*, and *PBRM1*, which have been described to be recurrently mutated in PDAC by the Cancer Genome Atlas Research Network [33], but none of these genes were found to be mutated by WES. However, when analyzing the copy number variations (CNV) of these genes, we detected losses of *KRAS*, *BRCA2*, *CDKN2A*, and *DPC4* as well as gains of *RNF43*, *GNAS,* and *PBRM1* in both cell lines. For *ARID1A*, we detected a loss in LuPanc-1 but a gain in LuPanc-2. Notably, a complete loss of *DPC4* (absence of exons 1–5, as confirmed by sequencing) and a loss of *RREB1* were found in LuPanc-2 but not LuPanc-1 cells, while, conversely, LuPanc-1 but not LuPanc-2 cells showed a loss of *TP53* and a gain of *CREBBP*. Figure 3B summarizes all mutations detected in both cell lines as a complete oncoplot. As we did not detect sufficient coverage for the *KRAS* codons 12 and 13 in the WES data, we specifically analyzed these loci by subsequent Sanger sequencing and detected a codon 12 mutation (Gly-Val) in both cell lines.

### 3.5. Sphere-Forming Capacity in Semi-Solid Medium

The mutational signatures and, in particular, the presence of mutant KRAS suggested that both cell lines possess a transformed phenotype. To scrutinize this functionally, we performed sphere formation assays, exploiting the ability of cancer cells to grow and generate visible colonies (spheres) from single-cells when cultured in a semi-solid medium. As expected, both cell lines generated large numbers of spheres in soft agar and Matrigel, with LuPanc-2 cells forming substantially more spheres than LuPanc-1 cells (Figure 4).

### 3.6. Characterization of TGF-β Signaling

Signaling by TGF-β plays antagonistic roles during carcinogenesis by initially inhibiting epithelial growth and later promoting the progression of advanced tumors, i.e., by stimulating migration/invasion, thus having a dual role as a tumor suppressor and promoter. Genes that define components of this pathway, such as *DPC4* (encoding SMAD4) and *TGFBR2* (encoding the type II receptor), have been described to be recurrently mutated in PDAC and to be associated with increased or decreased resistance to gemcitabine [14,15,16]. Moreover, inhibitors of the ALK5 kinase augmented the cytotoxicity of gemcitabine in both parental and gemcitabine-resistant pancreatic cancer cells [34], and Galunisertib, the first-in-class, first-in-human, oral small-molecule ALK5 inhibitor that enters clinical development in combination with gemcitabine, improved overall survival vs. gemcitabine alone in patients with unresectable pancreatic cancer [35]. In view of the gemcitabine viability assays shown below (see Section 3.7), it was of interest to evaluate the cells’ sensitivity to treatment with TGF-β. Towards this end, we treated cultures of LuPanc-1 and LuPanc-2 cells with rhTGF-β1 for 2 h. This induced C-terminal phosphorylation (reflecting activation) of the receptor-regulated Smad protein, SMAD2 (Figure 5A), suggesting the presence of functional TGF-β receptor signaling in both cell lines. To evaluate if signals transmit to the nucleus to result in corresponding changes in gene expression, we monitored several known TGF-β target genes for their response to rhTGF-β1. We found that both short-term (24 or 48 h) and long-term (15 d) stimulation of LuPanc-1 cells, but not LuPanc-2 cells, resulted in the upregulation of *SERPINE1* (encoding plasminogen activator-inhibitor type 1, PAI-1), *SNAI2* (encoding SNAIL2/SLUG), or *SOX2*, respectively (Figure 5B). In search of a molecular explanation for the unresponsiveness of LuPanc-2 cells to rhTGF-β1 stimulation with respect to changes in target gene expression, we recalled that, in WES analysis, *DPC4* was found to be lost in LuPanc-2 but not LuPanc-1 cells (Figure 3B). The absence of SMAD4 mRNA in LuPanc-2 cells was then confirmed by qPCR, while the levels in LuPanc1 were comparable to those in PANC-1 cells (Figure 5C) known to harbor wild-type *DPC4* and to be TGF-β-sensitive [19,20]. However, the abundance of mRNA for SMAD2 was equal in all three lines (Figure 5C), which is consistent with the presence of the SMAD2 protein (Figure 5A). From these data, it can be concluded that the failure of LuPanc-2 cells to respond to treatment with rhTGF-β1 is due to the absence of the common mediator Smad, SMAD4.

### 3.7. Migratory Capacity

We next evaluated the migratory capacity of LuPanc-1 and LuPanc-2 cells using the xCELLigence RTCA technology (OLS, Bremen, Germany), which represents a non-invasive and label-free approach for the continuous (real-time) monitoring of cell migration and invasion on a cell culture level [36]. For LuPanc-1 cells, high migratory activity was recorded, while that of LuPanc-2 cells was close to background levels (Figure 6A).

Having shown above that LuPanc-1 cells have retained sensitivity to rhTGF-β1 with respect to the activation of Smad proteins and invasion-associated target genes, we sought to know whether rhTGF-β1 would also impact the cells’ migratory behavior in vitro. To this end, this growth factor increased the migratory activity over that of control cells (blue curve/tracing B vs. green curve/tracing A, Figure 6B). Notably, the TGF-β type I receptor/ALK5 serine/threonine kinase inhibitor SB431542 was able to relieve this stimulatory effect (red curve/tracing C, Figure 6B), indicating that the TGF-β1 effect on migratory activity was specific and receptor-mediated. As expected, LuPanc-2 cells failed to respond to TGF-β1 stimulation with (enhanced) migratory activity under the same conditions (data not shown).

### 3.8. Treatment Response to Gemcitabine and FOLFIRINOX

As described above, gemcitabine and FOLFIRINOX are essential chemotherapeutics used in clinical routine practice for PDAC treatment. Accordingly, the sensitivity of both cell lines was tested in Resazurin-based viability assays for gemcitabine monotherapy and FOLFIRINOX by determining the respective half-maximal inhibition concentration (IC_50_). The IC_50_ for gemcitabine was 13.6 nM (IC_90_ of 50.5 nM) for LuPanc-1 and 357.1 nM for LuPanc-2 (IC_90_ of 2812 nM) (Figure 7, left-hand graph). The different levels of resistance of LuPanc-1 and LuPanc-2 cells to gemcitabine were verified by FACS analyses in independent experiments (Appendix A). The IC_50_ value for FOLFIRINOX (as a concentration of 5-FU in FOLFIRINOX) was 573.8 nM (IC_90_ of 5659 nM) for LuPanc-1 (Figure 7, right-hand graph). For LuPanc-2 cells, it was not possible to determine the IC_50_ (or IC_90_) for FOLFIRINOX since we were unable to obtain a normal dose–response curve.

## 4. Discussion

PDAC is a highly malignant carcinoma with an exceedingly poor prognosis. The overall 5-year survival rate of PDAC patients is as low as 5–8% [9]. Surgical resection of the tumor is the only curative treatment option, and patients usually receive adjuvant chemotherapy after surgery [37,38]. However, about 60% of the patients already progressed to metastatic disease at the time of diagnosis [39]. For these patients, palliative chemotherapy is the only treatment option. In both cases, patients are commonly treated using either mFOLFIRINOX or gemcitabine (with nab-paclitaxel), according to current clinical guidelines. The tumor biology of individual patients is not considered for the selection of the chemotherapeutic treatment regimen. Targeted therapies are currently not an integral part of the clinical treatment in PDAC patients [40].

Preclinical studies using patient-derived cell lines may be useful in predicting the response of a particular patient to gemcitabine/FOLFIRINOX or for the development of alternative and novel treatment strategies. Several studies have described specific molecular subtypes of PDAC based on transcriptional profiles [10,11,12]. The quasi-mesenchymal subtype by Collisson et al. [10], the basal subtype by Moffitt et al. [11] and the squamous subtype by Bailey et al. [12] significantly overlap and have a worse clinical outcome when compared to other subtypes [41,42]. The Cancer Cell Line Encyclopedia (CCLE) from the Broad Institute comprises 60 pancreatic cell lines derived from primary tumors or metastasis. However, these cell lines do not fully reflect the diverse and complex tumor biology of PDACs and clinical annotation; especially, the oncological outcome of the respective patient is mostly missing. The correlation of findings on primary ex vivo cultures with the clinical outcome of the individual patients is crucial in the era of personalized precision medicine.

In this study, we established two novel PDAC-derived cell lines named LuPanc-1 and LuPanc-2 from the surgical specimens of PDAC patients. Both cell lines presented with a transformed phenotype, as evidenced by their mutational spectra and their ability to form spheres in semi-solid media, also suggesting that the cells have stem cell properties. Clinical follow-up was annotated for both patients, and both corresponding cell lines were characterized comprehensively at the molecular level and functionally with respect to invasive behavior and the response to chemotherapeutic drugs.

Primary, patient-derived cell cultures are a highly relevant tool because they reflect the in vivo state and physiology better compared to immortalized secondary cell lines. Both cell lines established in this study were taken into the primary culture directly from surgical specimens. They can be maintained in the culture without changes in cell dynamics and a loss of viability, obviating the need for cell immortalization [43,44]. Most available cell lines have been cultured and used as workhorses for experimental cancer research for decades, while LuPanc-1 and LuPanc-2 are two low-passage cell lines. The clonal dynamics of cell cultures that have been passaged in various laboratories over extensive periods of time need to be considered during experimental design and data evaluation and, in some cases, may complicate data interpretation and comparability. We therefore presume that our novel cell lines are a highly valuable tool, in addition to the commonly used cell lines. Another unique feature is their full clinical annotation; most of the commercially available cell lines lack any clinical information. Combining the information of the cells’ biology with the clinical course of the patients might help in better understanding or even predicting treatment responses.

Finally, one of the main drawbacks in the design of personalized treatment strategies is the marked inter-tumor heterogeneity in PDAC. As mentioned above, although the CCLE database already contains 60 pancreatic cancer cell lines, these include histological entities other than PDAC, and those within the PDAC fraction are likely to not represent the full spectrum of PDAC histomorphological subtypes. Moreover, a substantial number of those cell lines have been derived from metastases rather than the primary tumor, which means that these do not necessarily respond in the same way to chemotherapeutic drugs as the cancer cells from the patient’s primary tumor. With respect to histomorphological differentiation, LuPanc-1 cells present with a mixed phenotype co-expressing both epithelial and mesenchymal markers at a single-cell level, as demonstrated for E-cadherin and vimentin by the immunofluoresecence and immunoblot analysis of single-cell-derived clonal cultures. This phenotype has so far only been described for the PANC-1 cell line and has been shown to reflect the presence of subpopulations of cells with different EMT phenotypes [18]. This mixed phenotype of LuPanc-1 cells is indicative of cells that have undergone hybrid or partial EMT. Notably, these cells have been shown to be more plastic and to exhibit higher tumorigenic and stemness potential, eventually resulting in enhanced metastasis and therapy resistance [45]. Therefore, LuPanc-1 cells might be particularly useful when analyzing these properties, as previously shown for PANC-1 cells [31,46,47,48].

We also determined the response of both cell lines to gemcitabine and FOLFIRINOX, which are the relevant treatment regimens in current clinical care. LuPanc-2 was substantially more resistant to gemcitabine (IC_50_ 357.1 nM) compared to LuPanc-1 (IC_50_ 13.6 nM). LuPanc-1 showed a similar response to gemcitabine compared to other commonly used cell lines such as CAPAN-1 (11.5 nM), BxPC-3 (18 nM), MIA PaCa-2 (36–40 nM), and PANC-1 (50 nM) [49,50,51,52]. The response of LuPanc-1 to FOLFIRINOX was in line with those of other cell lines published recently by Begg et al. [53]. Surprisingly, we were not able to obtain a typical dose response for the slowly growing LuPanc-2 line under treatment for FOLFIRINOX, which most likely reflects the high level of resistance. In line with the poorer histopathological differentiation of LuPanc-2 in comparison to that of LuPanc-1, LuPanc-2 showed a worse response to gemcitabine and FOLFIRINOX.

Both patients who gave rise to our cell lines received additive therapy with FOLFIRINOX after surgery but developed a local recurrence in a short postoperative interval of 9 and 8 months, respectively. The patient from whom LuPanc-1 was derived received gemcitabine treatment for the local recurrence, which is in line with a good response of the derived cell line in vitro. However, the detected *BRCA2* nonsense mutation might offer a rationale for an alternative treatment strategy based on PARP inhibitors such as olaparib.

The epithelial-mesenchymal phenotyping of both cell lines showed that while the phenotype of LuPanc-2 cells was tentatively epithelial based on the presence of CK7 and E-cadherin, along with the absence of vimentin protein expression, LuPanc-1 cells combined both epithelial and mesenchymal traits, revealing a mixed E/M phenotype. Cells that co-express markers of both subtypes have been postulated to possess a higher stem cell potential and cellular plasticity. Tumor cells with this mixed phenotype, which is also referred to as partial or hybrid EMT, have been shown to be more metastatic [54], which is consistent with the high migratory activity of LuPanc-1 cells in real-time cell migration assays (Figure 6). For instance, we have recently shown that PANC-1 cells, which also co-express epithelial and mesenchymal markers, have a high propensity for transdifferentiation into specialized cell types such as insulin-expressing cells in vitro [47], and they are highly invasive in vitro and in vivo [19,20]. It will therefore be exciting to reveal whether LuPanc-1 cells, too, are more prone to ductal-to-endocrine differentiation and if they exhibit greater metastatic activity in vivo in mouse models of PDAC.

Of particular interest was the finding of the deletion of *DPC4* in LuPanc-2 cells by CNV analysis, which was subsequently confirmed by SMAD4-specific qPCR. *DPC4* has been described to be recurrently mutated in PDAC, and its genomic deletion was significantly correlated positively with disease progression [55] and negatively with progression-free survival [56]. In addition, the loss of *DPC4* was associated with an altered sensitivity to treatment with gemcitabine [14,15,57]. A previous study using a genetically defined pancreatic cancer cell panel, which tested the cancer cells’ responses to chemotherapeutic drugs, found that SMAD4-inactivated cancer cell lines were more sensitive to cisplatin and irinotecan by two- to fourfold, but they were only modestly less sensitive to gemcitabine [58]. Despite the lack of a correlation between the *DPC4* status and the gemcitabine response in this panel of PDAC cell lines, among *DPC4* isogenic cell lines, *DPC4*-defective cells were about twofold less sensitive to gemcitabine than wildtype cells. A correlation analysis for IC_50_ values and TGF-β pathway activities in these isogenic cells confirmed a negative correlation with TGF-β pathway activities [58], matching the results from our study, in which a lack of TGF-β1 target gene responses correlated with a high gemcitabine IC_50_ in LuPanc-2 cells. However, it should be mentioned that, in another study, both SMAD4-deleted PDAC cell lines and patient-derived organoids exhibited a *higher* sensitivity toward gemcitabine (and other cell cycle-targeting agents) due to the upregulation of cell cycle-related genes [16], suggesting that—unlike in LuPanc-2—SMAD4 deletion may also result in the activation of mitosis. Moreover, pancreatic cancer cells with p16^INK4^ inactivation were three- to fourfold less sensitive to gemcitabine [58]. Since LuPanc-2 harbors a missense mutation (in addition to a loss) in *CDKN2A*, we speculate that the combined alterations in *CDKN2A* and *DPC4* are causal for the high gemcitabine IC_50_ of LuPanc-2 cells. Another interesting issue is whether the inability of LuPanc-2 cells to exhibit basal migratory activity in vitro is a direct consequence of their unresponsiveness to TGF-β1 or the SMAD4 loss, i.e., due to the failure of autocrine TGF-β to stimulate cell motility.

Cell lines and, in particular, primary patient-derived cell lines are an essential tool for preclinical drug stratification and can help in developing personalized treatment regimens [59,60]. However, a limitation of cell line models is the lack of a specific tumor microenvironment, which not only mechanically compromises drug delivery but also possesses immune and drug-scavenging activity [61,62,63]. When using our novel cell lines for preclinical drug testing, this limitation needs to be considered and possibly addressed using complementary techniques.

## Figures and Tables

**Figure 1 cells-12-00587-f001:**
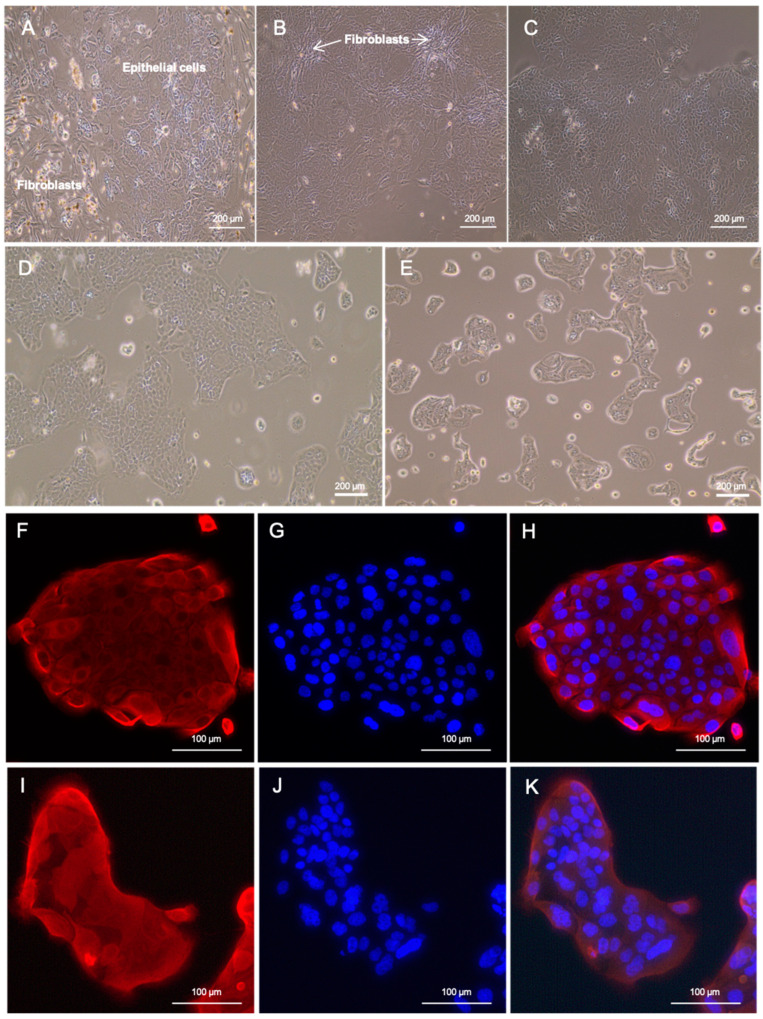
Morphology and immunofluorescent staining for CK7 in the patient-derived cell lines, LuPanc-1 and LuPanc-2. (**A**) LuPanc-1 cells 7 d after cell isolation. (**B**) LuPanc-1 cells after 30 d in culture and partial depletion of fibroblasts. (**C**) LuPanc-1 cells after total fibroblast depletion. (**D**) LuPanc-1 and (**E**) LuPanc-2 cells without fibroblasts. (**F**–**K**) Immunofluorescent staining of LuPanc-1 (**F**–**H**) and LuPanc-2 (**I**–**K**) cells for CK7 (**F**,**I**), DAPI (**G**,**J**), and overlay of CK7 and DAPI (**H**,**K**).

**Figure 2 cells-12-00587-f002:**
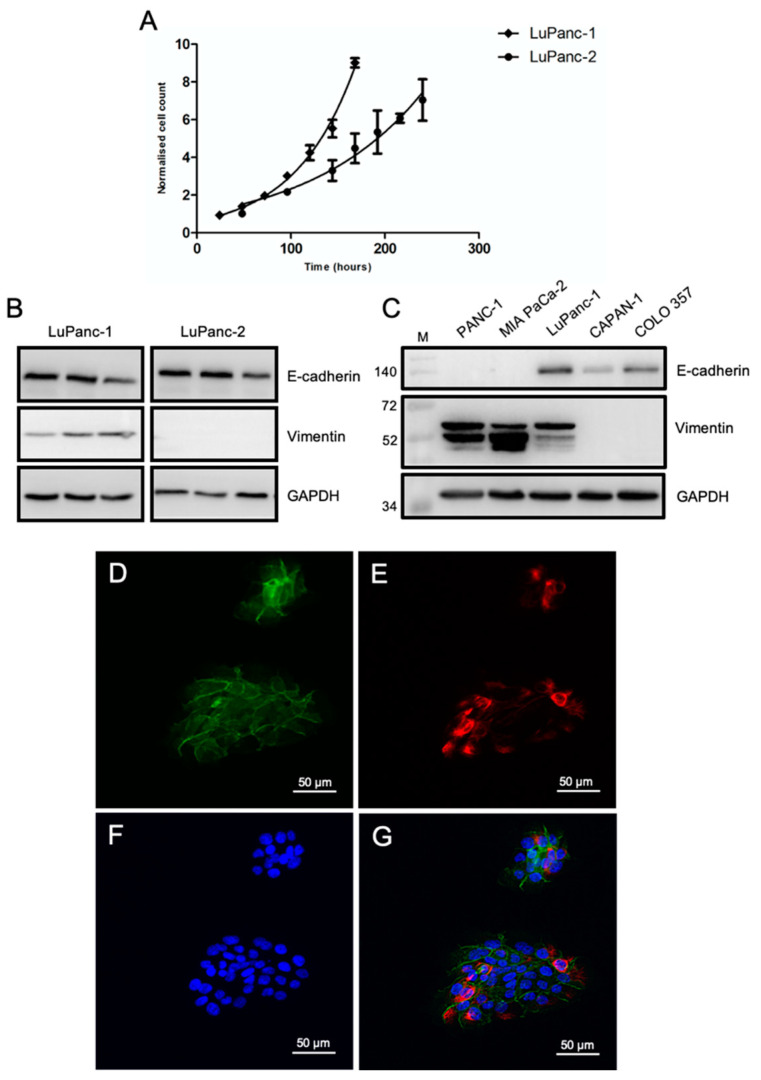
Molecular characterization of the LuPanc-1 and LuPanc-2 cell lines. (**A**) Growth curves for LuPanc-1 and LuPanc-2. (**B**) Crude proteinaceous extracts were prepared from both cell lines and subjected to the immunoblot analysis of E-cadherin and vimentin. (**C**) As the control, E-cadherin and vimentin expression in LuPanc-1 cells is shown in comparison to that in other PDAC-derived tumor cell lines of either the quasimesenchymal (PANC-1, MIA PaCa-2) or epithelial (CAPAN-1, COLO 357) subtype [31]. The detection of GAPDH was used in (**B**) and (**C**) to verify equal loading. (**D**–**G**) Double immunofluorescence staining of E-cadherin (**D**) and vimentin (**E**) in small clusters of LuPanc-1 cells. Nuclei were counterstained with DAPI (**F**), and concurrent expression was visualized by overlay (**G**).

**Figure 3 cells-12-00587-f003:**
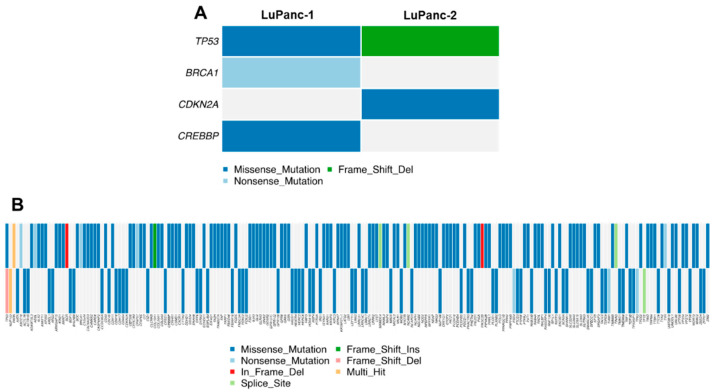
Mutational landscape of LuPanc-1 and LuPanc-2 cells based on WES. (**A**) Mutations in common oncogenes and tumor suppressor genes detected in both cell lines, according to Vogelstein and colleagues [32]. (**B**) Complete oncoplot of all mutations detected in both cell lines. For an enlargement of this oncoplot, see Appendix A.

**Figure 4 cells-12-00587-f004:**
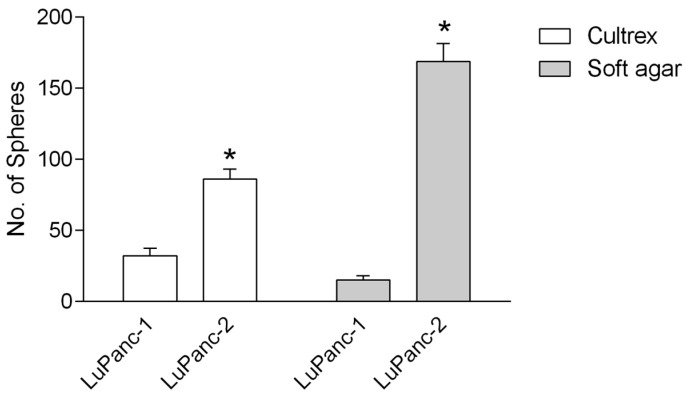
Tumor sphere assay of LuPanc-1 and LuPanc-2 cells. Both cell lines formed spheres in cultrex and soft agar. LuPanc-2 cells formed significantly more spheres compared to LuPanc-1 cells under both conditions. Data are the mean ± SD of two biological replicates, each comprising three technical triplicates. The asterisks (*) indicate a significant difference (*p* < 0.05, Student’s *t* test).

**Figure 5 cells-12-00587-f005:**
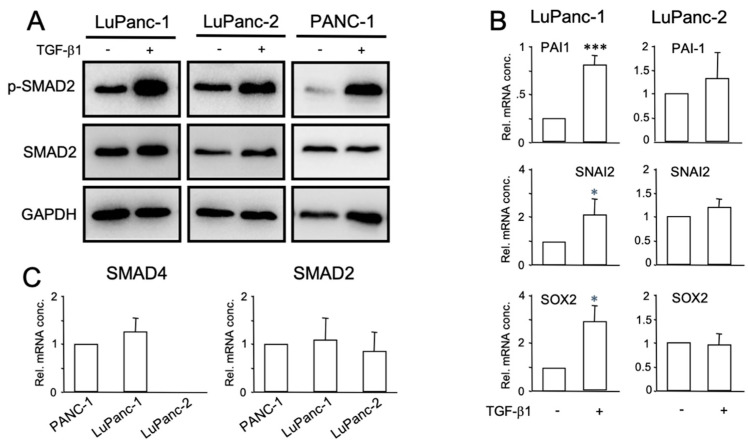
Characterization of TGF-β signaling in LuPanc-1 and LuPanc-2 cells. (**A**) Phospho-immunoblotting of SMAD2 in LuPanc-1, LuPanc-2, and PANC-1 cells as the control. Cells were treated with rhTGF-β1 for 2 h and subjected to sequential immunoblotting of phospho-SMAD2 (p-SMAD2), (total) SMAD2, and GAPDH to control for equal loading. (**B**) Monitoring of TGF-β target genes for their response to treatment with rhTGF-β1. LuPanc-1 and LuPanc-2 cells were stimulated short-term (48 h) for PAI-1 and SLUG or long-term (15 d) for SOX2 with rhTGF-β1 (5 ng/mL) and subjected to qPCR for *SERPINE1*, *SNAI2*, or *SOX2*, respectively. The asterisks (∗) indicate a significant difference; ∗ *p* < 0.05, ∗∗∗ *p* < 0.001. (**C**) Detection of SMAD4 mRNA levels by qPCR in LuPanc-1 and LuPanc-2 cells relative to those in PANC-1 cells set arbitrarily at 1.0. SMAD2 mRNA was detected in the same samples as a positive control.

**Figure 6 cells-12-00587-f006:**
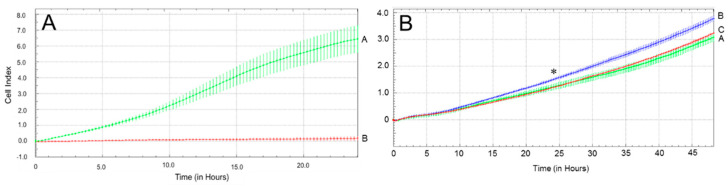
Measurement of basal and TGF-β-induced migratory capacity of LuPanc-1 and LuPanc-2 cells. (**A**) Both cell lines (green curve/tracing A: LuPanc-1, red curve/tracing B: LuPanc-2) were subjected to a real-time cell migration assay on an xCELLigence platform for 24 h. The relative migratory activity is plotted on the ordinate and given as the dimensionless Cell Index. Data are the mean ± SD of quadruplicate wells. The assay shown is representative of three assays performed in total. (**B**) The same procedure as that in (**A**), except that LuPanc-1 cells were assayed in the absence (green curve/tracing A) or presence (blue curve/tracing B) of rhTGF-β1 (5 ng/mL) for 48 h. In another four wells, cells were treated with rhTGF-β1 plus the ALK5 serine/threonine kinase inhibitor SB431542 (5 µM, red curve/tracing C). Data shown are the mean ± SD of quadruplicate wells from one out of two assays. The asterisk (∗) marks the time point at which the difference between TGF-β1-treated and untreated control cells was first statistically significant. Differences remained as such until the end of the assay. No significant differences were noted between the untreated control and the rhTGF-β1 + SB431542-treated cells.

**Figure 7 cells-12-00587-f007:**
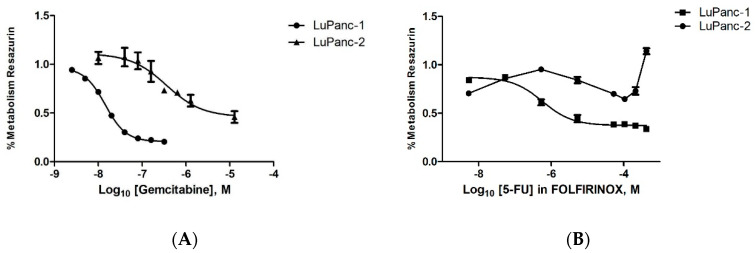
Evaluation of the proliferative response of LuPanc-1 and LuPanc-2 cells to gemcitabine or FOLFIRINOX. Both cell lines were treated with either gemcitabine (**A**) or FOLFIRINOX (**B**), followed by the addition of Resazurin and OD measurement.

**Table 1 cells-12-00587-t001:** Short tandem repeat analysis.

Locus	LuPanc-1	LuPanc-2
D3S1358	16/17	15,17
D1S1656	11/14	12,15
D6S1043	12/17	19
D13S317	11/12	11/12
Penta E	11	7/12
D16S539	11/13	12,13
D18S51	16	13
D2S1338	22/23	17/23
SCF1PO	10/12	10/11
Penta D	13/15	10
TH01	7/9	6/9
vWA	17/20	14/19
D21S11	30	27/30
D7S820	11	8/10
D5S818	11/13	13
TPOX	12	8
D8S1179	12/13	13/15
D12S391	21/23	17/19
D19S433	14/15	15
FGA	22/24	20/21.2
AM	X/Y	X

**Table 2 cells-12-00587-t002:** Total number of mutations after processing.

	Frame Shift Deletion	Frame Shift Insertion	In-Frame Deletion	Missense Mutation	Nonsense Mutation	Splice Site	Total
LuPanc-1	0	1	2	124	5	3	135
LuPanc-2	1	0	0	70	4	1	76

## Data Availability

All data reported are contained in the “Results” section and the Appendix A.

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
