# Peer review of "Establishment and Molecular Characterization of Two Patient-Derived Pancreatic Ductal Adenocarcinoma Cell Lines as Preclinical Models for Treatment Response"

_cells, 2023, doi:10.3390/cells12040587_

Round 1

Reviewer 1 Report (Previous Reviewer 2)

In their manuscript entitled “Establishment and molecular characterization of two patient-derived pancreatic ductal adenocarcinoma cell lines as preclinical models for treatment response”, Braun et al present a brief characterization of two new PDAC cell lines they created. The article is well written, and describes new models that could be of potential interest for researchers working on PDAC. While more in-depth studies will be required in order to better characterize these new cell lines, I recommend the publication of this manuscript after my comments are addressed.

Minor comments:

Line 2: “adenocarcinoma” instead of “adenocarinoma”.

Lines 264-267: “metabolic activity was substantially slower in LuPanc-2”… “this effect is caused by higher metabolic activity of LuPanc-2 cells”. These statements are in apparent contradiction. Could the authors please correct if needed, and elaborate on these results? Indeed, the lack of difference between LuPanc-1 and LuPanc-2 cells at the last time point (Figure S1) is surprising, and weakens the authors’ conclusions based on this assay.

Figure 4: The spelling in the figures indicates “LüPanc” instead of “LuPanc” everywhere else in the manuscript.

Supplementary figure 2 is blank.

Author Response

Dear Mrs. Guan,

We thank you for your email dated December 21st 2022. We highly appreciate your willingness to consider a revised version of our manuscript for publication in Cells. We would also like to thank the reviewers for their helpful comments, which we addressed below point-by-point and which we think have considerably enhanced the quality of our manuscript.

References are provided below for each point where necessary. All changes to the previous version have been highlighted in the “track changes" mode for easier visibility.

Reviewer 1

Comment 1:

In their manuscript entitled “Establishment and molecular characterization of two patient-derived pancreatic ductal adenocarcinoma cell lines as preclinical models for treatment response”, Braun et al present a brief characterization of two new PDAC cell lines they created. The article is well written, and describes new models that could be of potential interest for researchers working on PDAC. While more in-depth studies will be required in order to better characterize these new cell lines, I recommend the publication of this manuscript after my comments are addressed.

Response: We highly appreciate the comments of Reviewer 1. As described below, we have performed additional experiments to better characterize our new PDAC cell lines. All comments are addressed below. Please note that we have also added novel data in response to requests from Reviewer 3. When considering all these new data, we believe that an in-depth characterization of our cell lines has now been achieved. 

Comment 2:

Line 2: “adenocarcinoma” instead of “adenocarinoma”.

Response: We thank reviewer 1 and have corrected this error.

Comment 3:

Lines 264-267: “metabolic activity was substantially slower in LuPanc-2”… “this effect is caused by higher metabolic activity of LuPanc-2 cells”. These statements are in apparent contradiction. Could the authors please correct if needed, and elaborate on these results? Indeed, the lack of difference between LuPanc-1 and LuPanc-2 cells at the last time point (Figure S1) is surprising, and weakens the authors’ conclusions based on this assay.

Response: We agree with Reviewer 1 that these statements were contradictory. The considerable increase in metabolic activity between the 96 and 120 h time points was reproduced in three independent experiments, each with three technical replicates. Our assumption that this effect was caused by higher metabolic activity of LuPanc-2 cells – which, in turn, may be due to the larger cell size compared to LuPanc-1 - was admittedly speculative. We have therefore removed this statement.

Comment 4:

Figure 4: The spelling in the figures indicates “LüPanc” instead of “LuPanc” everywhere else in the manuscript.

Response: We thank the reviewer and have corrected “LüPanc” to “LuPanc” in Figure 4 (now Figure 5 in the revised version).

Comment 5:

Supplementary figure 2 is blank.

Response: We have added the former Supplementary Figure S2 (now Figure S3) to the Supplementary data file.

Reviewer 2 Report (New Reviewer)

This study aimed to investigate characteristics of tumor in two novel patient-derived primary cell lines, LuPanc-1 and LuPanc-2 and to determine tested to standard therapy regimens gemcitabine and FOLFIRINOX. I have several comments for the authors’ consideration to further improve the manuscript.

1. On line 267, [which may be due to the larger cell size compared to Lu-Panc-1 cells (Figure 1D&E)]-> In Figure 1. (D) is the morphology of LuPanc-1 and (E) is the morphology of LuPanc-2. Lu-Panc-1 cells appear larger than Lu-Panc-2 cells, which needs confirmation. You need a cell size quantification.

2. To evaluate the treatment of gemcitabine and FOLFIRINOX in LuPanc-1 and LuPanc-2, a xenograft model is needed.

Author Response

Dear Mrs. Guan,

We thank you for your email dated December 21st 2022. We highly appreciate your willingness to consider a revised version of our manuscript for publication in Cells. We would also like to thank the reviewers for their helpful comments, which we addressed below point-by-point and which we think have considerably enhanced the quality of our manuscript.

References are provided below for each point where necessary. All changes to the previous version have been highlighted in the “track changes" mode for easier visibility.

Reviewer 2

Comment 1:

On line 267, [which may be due to the larger cell size compared to Lu-Panc-1 cells (Figure 1D&E)]-> In Figure 1. (D) is the morphology of LuPanc-1 and (E) is the morphology of LuPanc-2. Lu-Panc-1 cells appear larger than Lu-Panc-2 cells, which needs confirmation. You need a cell size quantification.

Response: We thank Reviewer 2 for this comment. Accordingly, we aimed to quantify cell size for both cell lines. However, we were unable to clearly display cytoplasmic boundaries of single cells (see Figure 1B below), which prevented a reproducible cell size quantification. Hence, we agree with Reviewer 2 that our estimation of cell sizes was rather speculative and therefore have removed the respective statement (lines 267 f.).

Figure 1: Phase contrast images of LuPanc-1 (A) and LuPanc-2 (B) cells (see attached file).

Comment 2:

To evaluate the treatment of gemcitabine and FOLFIRINOX in LuPanc-1 and LuPanc-2, a xenograft model is needed.

Response: This is an excellent idea; however, this study was designed as an in-vitro study. We agree with Reviewer 2 that besides patient-derived cell lines and patient-derived organoids xenografts are excellent tools to evaluate treatment response. However, the focus of this study is on patient-derived primary cell lines. In our opinion, quantification of differential treatment responses of cell lines in vitro is well established in experimental cancer research. Nevertheless, we agree with Reviewer 2 that xenograft experiments might be of interest in the future but are beyond the scope of this study.

Reviewer 3 Report (New Reviewer)

The paper by Braun et al tackles a very important gap in Pancreatic cancer research field: lack of relevant and adequate pre-clinical models. Having said so, the paper can be significantly modified to make it more scientifically appealing. Please find my comments below:

1- Please revisit the English language because there are many grammatical mistakes and unclarity.

2- Please remove any highlight from text as it is distracting

3- In Figure 1, CK7 is of bad quality. Please provide better images and zoom in if possible.

4- The fact that LuPanc-1 expresses both Vim and E-Cadherin could be very interesting. do we know if we have 2 different populations of cells where one expresses E-cad and the other expresses Vim? this can be tackled by performing a double Immunofluorescent analysis of Vim/Ecad on the cells, which is an important step for the validation of the cell model. In addition, one possibility could be contamination with fibroblast (despite the fact that magnetic sorting was done). to rule out the latter, you can either sort the cells for E-cad positive and negative and culture them and reassess if you have both Vim/Ecad stain. Altrenatively, you can clone the cells and pick independant colonies and check for Vim/Ecad expression. It is important to know if you have contamination or you have spontaneous EMT taking place.

4- Do we know if the novel cells have stem cell properties? a tumorsphere assay can be performed to check and it adds value to the model.

5- Are the cells tumorigenic?

Author Response

Dear Mrs. Guan,

We thank you for your email dated December 21st 2022. We highly appreciate your willingness to consider a revised version of our manuscript for publication in Cells. We would also like to thank the reviewers for their helpful comments, which we addressed below point-by-point and which we think have considerably enhanced the quality of our manuscript.

References are provided below for each point where necessary. All changes to the previous version have been highlighted in the “track changes" mode for easier visibility.

Reviewer 3

Comment 1:

The paper by Braun et al tackles a very important gap in Pancreatic cancer research field: lack of relevant and adequate pre-clinical models. Having said so, the paper can be significantly modified to make it more scientifically appealing. Please find my comments below:

Response: We thank Reviewer 3 for appreciating our manuscript as tackling a very important gap in the field of pancreatic cancer research. Accordingly, we have addressed all comments made by Reviewer 3 and performed additional experiments. When considering all the new data added in response to the three reviewers, we believe that the scientific appeal of our manuscript has increased significantly. 

Comment 2:

Please revisit the English language because there are many grammatical mistakes and unclarity.

Response: Our manuscript was revised linguistically by a native speaker (Darawalee Wangsa Zong, PhD), whose name was added to the Acknowledgement section.

Comment 3:

Please remove any highlight from text as it is distracting.

Response: We have removed all colored highlighting from our previous revision and highlighted - by the “track changes” mode - changes made for the current revision according to the publisher’s requirements.

Comment 4:

In Figure 1, CK7 is of bad quality. Please provide better images and zoom in if possible.

Response: We have replaced the images of the immunofluorescent stains for CK7 in panels F-K of Figure 1. We have inserted clearer images with higher magnification.

Comment 5:

The fact that LuPanc-1 expresses both Vim and E-Cadherin could be very interesting. do we know if we have 2 different populations of cells where one expresses E-cad and the other expresses Vim? this can be tackled by performing a double Immunofluorescent analysis of Vim/Ecad on the cells, which is an important step for the validation of the cell model. In addition, one possibility could be contamination with fibroblast (despite the fact that magnetic sorting was done). to rule out the latter, you can either sort the cells for E-cad positive and negative and culture them and reassess if you have both Vim/Ecad stain. Altrenatively, you can clone the cells and pick independant colonies and check for Vim/Ecad expression. It is important to know if you have contamination or you have spontaneous EMT taking place.

Response: We thank reviewer 3 for this very valuable suggestion to further evaluate the expression of Ecad and Vim in LuPanc-1. Accordingly, we have applied two of the three experimental strategies suggested. Firstly, we performed a double immunofluorescent staining of Ecad and Vim of LuPanc-1 cells (Figure 2, panels D-G). Interestingly, these stains show that LuPanc-1 cells indeed co-express both proteins but are highly heterogenous in the strength of Ecad and Vim expression. Secondly, we established by the technique of limited dilution clonal lines derived from single cells and monitored these for Ecad and Vim expression by immunoblotting. Again, it was evident that all six clones analyzed differed in their expression of both Ecad and Vim with differences among the clones being stronger for Vim than for Ecad. However, the ratio of Ecad to Vim was similar among the clones, except for one clone (6F9), which displayed very strong Vim expression and hence a low Ecad:Vim ratio.

Comment 6:

Do we know if the novel cells have stem cell properties? a tumorsphere assay can be performed to check and it adds value to the model.

Response: Again, we thank Reviewer 3 for this highly valuable suggestion. As suggested, we have performed for both cell lines sphere formation assays in semi-solid media (Cultrex and soft agar). Intriguingly, both lines possess the ability to form tumor spheres, indicating that they have a transformed phenotype and possess stem cell properties. These data have been included in the revised version as new Figure 4. The numbering of all subsequent sections and figures has been changed accordingly.

Comment 7:

Are the cells tumorigenic?

Response: To test the tumorigenicity of our cell lines in an orthotopic mouse model of PDAC is an excellent idea, however, this study was designed as an in-vitro study hence the animal experiments were beyond the scope of it. However, we nevertheless thank Reviewer 3 for this highly valuable suggestion and we will consider in vivo experiments for future follow-up studies.

Round 2

Reviewer 2 Report (New Reviewer)

The author faithfully performed the requested comments. This thesis will be helpful to many researchers in this field.

Reviewer 3 Report (New Reviewer)

The Authors have addressed all my questions and concerns and I would move to accept this paper.

This manuscript is a resubmission of an earlier submission. The following is a list of the peer review reports and author responses from that submission.

Round 1

Reviewer 1 Report

The authors characterized at cellula and molecular level two PDAC cell lines derived from patients.

The data sound good and the experiments were well performed.

However, I have some comments regarding the data:

1) Fig.2 : Why does vimentin W.B. show two bands in panel C and one in panel B?

2) Fig. 4: The SMAD2/SMAD2p ratio showed an increase of SMAD2p after stimulation with TGF-beta, and SMAD2 is always higher than SMAD2p in almost all the analysis but not in LuPanc-1 + sample. Why? Moreover, the GAPDH was not evaluated for the relative quantification of total SMAD-2. 

3) Most of the papers that evaluated IC50 in PDAC cell lines described a GEM IC50 ranged from 490 nM to 24 microM, while the LuPanc-1 is very sensitive to GEM (13.6 nM). This data should be explained.
In addition, also the IC50 data for FOXFIRINOX are not in agreement with previous papers (for example FOLFIRINOX Versus Gemcitabine-based Therapy for Pancreatic Ductal Adenocarcinoma: Lessons from Patient-derived Cell Lines. Anticancer research 2020;

Reviewer 2 Report

In this study, Braun et al present a general overview of the key characteristics of 2 PDAC cells lines they have generated using human tumor samples. Both cell lines were sequenced, and mutations highlighted. Migration and proliferation were assessed, as well as the response to relevant chemotherapeutic drugs.

The manuscript is well written, and the 2 cell lines could prove to be of interest to the research community working on PDAC.

There is however no clear indication about how these cell lines differ from the already available 46 other cell lines mentioned by the authors. Do they really present new features or relevant markers? The authors only provide two figures (i.e., figures 2 and 4) where some comparison is proposed with respectively 4 and 1 commonly used PDAC cell lines, but this remains very preliminary, and some extra analysis is required before these new cell lines can be considered of interest. Moreover, proliferation, migration and survival are only assessed with one method: a more comprehensive analysis could help determining whether these cell lines present interesting features.